# OpenReview forum: "TimeStep Master: Asymmetrical Mixture of Timestep LoRA Experts for Versatile and Efficient Diffusion Models in Vision"
_ICML.cc/2025/Conference — ICML 2025 poster_

### Official Review · Reviewer_eNpx · 2025-03-08

**Overall Recommendation:** 3

**Summary:**

The paper introduces TimeStep Master (TSM), a novel approach for fine-tuning diffusion models efficiently. Unlike traditional Low-Rank Adaptation (LoRA), which applies the same tuning across all timesteps, TSM uses TimeStep LoRA experts specialized for different noise levels. It consists of two stages: fostering, where different LoRAs are trained for specific timestep intervals, and assembling, which combines these experts asymmetrically using a core-context mechanism. This method improves adaptability and achieves state-of-the-art results in domain adaptation, post-pretraining, and model distillation, demonstrating strong generalization across architectures and visual modalities.

**Claims And Evidence:**

Yes.

**Essential References Not Discussed:**

No.

**Experimental Designs Or Analyses:**

Yes

**Methods And Evaluation Criteria:**

Yes.

**Other Comments Or Suggestions:**

Too many tables are included in the main paper. I suggest moving in the supplementary material or simplifying some of them to free up space for more visualization results.

**Other Strengths And Weaknesses:**

Strength:
1. This paper introduces a general and concise TimeStep Master paradigm with two key fine-tuning stages. The fostering stage (1-stage), apply different LoRAs to fine-tune the diffusion model at different timestep intervals.  The assembling stage (2-stage) use a novel asymmetrical mixture of TimeStep LoRA experts, via core-context collaboration of experts at multi-scale intervals.
2. The extensive experiments are conducted on three typical and popular LoRA-related tasks of diffusion models, including domain
adaptation, post-pretraining, and model distillation.

Weakness:
1. Insufficient Visualization Results: The paper lacks sufficient visualization results, which are particularly critical for a study focusing on fine-tuning diffusion models. Adding more visual examples, such as qualitative comparisons, intermediate outputs, or generated results, would provide a clearer and more intuitive demonstration of the effectiveness of the proposed TSM paradigm. Visualizations should ideally be included in the main paper to ensure direct evidence of the method’s performance is presented upfront, rather than being relegated to the supplementary material.
2. The novelty is limited. While the proposed method appears to be relatively straightforward, essentially extending the original LoRA to different timesteps and assembling them. The lack of more complex or novel mechanisms may limit the perceived novelty of the approach. The authors should clarify how this simplicity contributes to the method’s practicality and scalability, or discuss potential directions for further enhancement.
3. The performance difference between TSM-Stage1 and TSM-Stage2 in Table 6 is very small. Including visual comparisons that highlight the contributions of each stage would help validate the effectiveness and necessity of the two-stage design. This would also provide a clearer understanding of how the assembling stage (Stage 2) augments the fostering stage (Stage 1).

**Questions For Authors:**

See weakness.

**Relation To Broader Scientific Literature:**

This paper advances the field of LORA fine-tuning by introducing the TimeStep Master (TSM) paradigm, which optimizes the application of diffusion models in tasks such as text-to-image/video generation (e.g., Flux and Sora). It underscores the necessity for efficient fine-tuning methods like LoRA to manage the extensive number of parameters involved, thereby enhancing both performance and computational efficiency.

**Theoretical Claims:**

Yes

---

> ### Author Rebuttal · Authors · 2025-03-28
>
> Response to Reviewer eNpx:
>
> We sincerely appreciate your thoughtful and detailed review. Your insightful comments have been invaluable in guiding us to improve our manuscript. Below we provide our point-by-point responses, and we hope our clarifications and planned enhancements address your concerns.
>
> **Question1:** : The paper lacks sufficient visualization results, which are particularly critical for a study focusing on fine-tuning diffusion models.
>
> **Answer1:** Thank you for your reminder. We realize that visualization is very important for generation tasks. Due to space constraints, we had to place some visualizations in the supplementary material, only showing a comparison example with traditional methods in the main paper teaser. We will add more visualizations to the main paper in the final version, including comparison results between TSM 1-stage and 2-stage to further demonstrate TSM's effectiveness and superiority, as well as visualization of intermediate layer outputs and generation results at different training steps to further illustrate the effectiveness of our method.
>
> ---
>
> **Question2:** The novelty is limited. While the proposed method appears to be relatively straightforward, essentially extending the original LoRA to different timesteps and assembling them. The lack of more complex or novel mechanisms may limit the perceived novelty of the approach. The authors should clarify how this simplicity contributes to the method’s practicality and scalability, or discuss potential directions for further enhancement.
>
> **Answer2:** Thank you for this perceptive comment. We appreciate the opportunity to elaborate on the innovative aspects of our approach. Despite its straightforward formulation, TSM includes several key contributions that we believe represent a significant advancement:
> 1. **Ensemble Strategy**: We cleverly ensemble LoRAs trained with different timestep intervals via gating mechanism to leverage complementary knowledge across intervals.
> 2. **Comprehensive Validation**: We comprehensively validate TSM across three architectures (UNet, DiT, MM-DiT), two modalities (image/video), and three task types (domain adaptation, post-pretraining, model distillation).
> 3. **Model Scalability**: We verify TSM's effectiveness on pretrained models of varying scales, from PixArt (0.6B) to SD3 (2B).
> We plan to extend experiments to more tasks like text-to-image generation, image colorization, and image restoration.
>
> Furthermore, we are excited to extend our experimental evaluations to include tasks such as text-to-image generation, image colorization, and image restoration. We truly appreciate your suggestion, which has motivated us to further clarify how the simplicity of our method contributes to both its practicality and scalability.
>
> ---
>
> **Question3:** The performance difference between TSM-Stage1 and TSM-Stage2 in Table 6 is very small.
>
> **Answer3:** Thank you for your keen observation. The performance difference observed in Table 6 specifically corresponds to distillation tasks. As indicated in Tables 4 and 5, the two-stage approach shows a more pronounced improvement for domain adaptation and post-pretraining tasks. We hypothesize that the relatively modest improvement in distillation may be due to early performance saturation after the first stage, leaving less room for further gains. We appreciate your detailed attention and will provide additional discussion on this aspect in the final revision.
>
> ---
>
> **Question4:** Including visual comparisons that highlight the contributions of each stage would help validate the effectiveness and necessity of the two-stage design. This would also provide a clearer understanding of how the assembling stage (Stage 2) augments the fostering stage (Stage 1).
>
> **Answer4:** We completely agree that visual comparisons can effectively highlight the contributions of each stage. In response, as mentioned in our answer to Question 1, we will include clear and side-by-side visual examples comparing TSM 1-stage and TSM 2-stage. This will better illustrate the overall impact and the incremental benefits brought by the second stage. Thank you for this excellent suggestion.
>
> ---
> Once again, we are truly grateful for your encouraging and incisive feedback. Your comments have inspired us to refine our manuscript further, and we hope that the planned revisions will enhance the clarity and impact of our work. Please do not hesitate to let us know if there are any additional details or clarifications that would be helpful.
>
> Thank you for your time and consideration.

---

### Official Review · Reviewer_zVmL · 2025-03-12

**Overall Recommendation:** 3

**Summary:**

This paper introduces TimeStep Master (TSM), a diffusion fine-tuning framework using an asymmetrical mixture of timestep LoRA experts. Rather than applying a single LoRA module across all timesteps, which limits the adaptability of different noise levels in the diffusion process, TSM introduces a timestep-specific fine-tuning approach. Specifically, the authors present an asymmetric mixture of LoRA experts, dividing the entire timestep into different intervals, then using the smallest-scale interval as the core expert and the rest as context experts. Core experts address fine-grained noise modeling while context experts are adaptively weighted based on the timesteps. Extensive experiments verify the effectiveness of TSM.

**Claims And Evidence:**

The authors build on existing evidence that diffusion models are trained to represent different data distributions at different timesteps, a concept previously framed as a multi-task learning problem. While this motivation is somewhat not novel, they strengthen their claim by constructing extensive experiments for various diffusion fine-tuning tasks.

**Essential References Not Discussed:**

Many other timestep-based diffusion training [1, 2, 3, 4] and fine-tuning methods [5, 6] are missing.

[1] Park et al., Denoising Task Routing for Diffusion Models, ICLR 2024. \
[2] Park et al., Switch Diffusion Transformer: Synergizing Denoising Tasks with Sparse Mixture-of-Experts, ECCV 2024. \
[3] Hatamizadeh et al., Diffit: Diffusion vision transformers for image generation, ECCV2024. \
[4] Choi et al., Simple Drop-in LoRA Conditioning on Attention Layers Will Improve Your Diffusion Model, TMLR 2024. \
[5] Fang et al., Remix-DiT: Mixing Diffusion Transformers for Multi-Expert Denoising, NeurIPS 2024. \
[6] Ham et al., Diffusion Model Patching via Mixture-of-Prompts, AAAI 2025.

**Experimental Designs Or Analyses:**

The authors use vanilla LoRA as the primary baseline to validate the effectiveness of TSM. However, since TSM incurs higher computational costs, its improved performance may be expected. To properly demonstrate the benefits of the asymmetric mixture of LoRA experts, the authors should either design an experimental setup with a similar computational cost to vanilla LoRA or conduct comparative experiments against other fine-tuning methods.

One potential experiment could involve extending the ablation studies on MoE LoRA to include other timestep-based LoRA conditioning methods [4] and alternative fine-tuning approaches [5, 6] (as referenced above).

**Methods And Evaluation Criteria:**

The authors validate the effectiveness of TSM across various diffusion fine-tuning tasks, but the analysis of its design choices is insufficient. Recent studies have provided in-depth analyses of timestep modeling in diffusion training, including approaches that divide timesteps into finer intervals to capture inter-task relationships between denoising tasks [1, 2] and methods that enhance timestep conditioning in attention layers [3, 4]. Building on these analyses, some works have proposed more detailed fine-tuning strategies based on timesteps [5, 6]. Therefore, the authors should support their claim by constructing some experiments to confirm compatibility or logical connections with prior timestep-based training and fine-tuning approaches.

[1] Park et al., Denoising Task Routing for Diffusion Models, ICLR 2024. \
[2] Park et al., Switch Diffusion Transformer: Synergizing Denoising Tasks with Sparse Mixture-of-Experts, ECCV 2024. \
[3] Hatamizadeh et al., Diffit: Diffusion vision transformers for image generation, ECCV2024. \
[4] Choi et al., Simple Drop-in LoRA Conditioning on Attention Layers Will Improve Your Diffusion Model, TMLR 2024. \
[5] Fang et al., Remix-DiT: Mixing Diffusion Transformers for Multi-Expert Denoising, NeurIPS 2024. \
[6] Ham et al., Diffusion Model Patching via Mixture-of-Prompts, AAAI 2025.

**Other Comments Or Suggestions:**

The claim would be more robust if the authors could validate that insights and observations from previous works are compatible with TSM.

**Other Strengths And Weaknesses:**

* **Strengths**: The authors provide comprehensive experiments to verify the effectiveness of TSM. Additionally, the asymmetric mixture of LoRA experts has a strong potential for scalable diffusion fine-tuning by increasing the number of context experts.

* **Weaknesses**: However, the paper lacks sufficient justification for the proposed method and comparative experiments to validate its superiority.

**Questions For Authors:**

Does TSM exhibit any scaling properties by increasing the number of intervals m? Many MoE-based diffusion training methods [1, 2] analyze their scaling behavior, so it would be valuable if the authors could provide insights into this aspect.

[1] Fei et al., Scaling Diffusion Transformers to 16 Billion Parameters, arXiv 2024. \
[2] Sun et al., EC-DIT: Scaling Diffusion Transformers with Adaptive Expert-Choice Routing, ICLR 2025.

**Relation To Broader Scientific Literature:**

Extensive experiments confirm that their approach aligns with observations in previous work and demonstrates effectiveness across various diffusion fine-tuning tasks. However, its superiority over existing methods remains unproven.

**Theoretical Claims:**

No theoretical claims and proofs.

---

> ### Author Rebuttal · Authors · 2025-03-28
>
> Response to Reviewer zVmL:
>
> Thank you for your careful reading and analysis of our article, and for providing valuable feedback. Below are our responses to your comments:
>
> **Question1:** Recent studies have provided in-depth analyses of timestep modeling in diffusion training, including approaches that divide timesteps into finer intervals to capture inter-task relationships between denoising tasks [1, 2] and methods that enhance timestep conditioning in attention layers [3,4]. Building on these analyses, some works have proposed more detailed fine-tuning strategies based on timesteps [5,6].
>
> **Answer1:**  We sincerely appreciate your reminder. We will supplement references to these works in the final version and provide detailed comparisons:
> - *Denoising Task Routing for Diffusion Models [1]* introduces channel mask strategies for different timesteps during training to inject temporal priors.
> - *Switch Diffusion Transformer: Synergizing Denoising Tasks with Sparse Mixture-of-Experts [2]* employs timestep-conditioned gating mechanisms to regulate expert activation.
> - *Diffit: Diffusion vision transformers for image generation [3]* adapts ViT architectures for image generation.
> - *Simple Drop-in LoRA Conditioning on Attention Layers Will Improve Your Diffusion Model [4]* designs multi-expert ensembles for class labels and timesteps.
>
> Above methods are typically used in pretraining.
>
> Methods like *Remix-DiT: Mixing Diffusion Transformers for Multi-Expert Denoising [5]* (learnable coefficients for timestep-specific fine-tuning) and *Diffusion Model Patching via Mixture-of-Prompts [6]* (gating modules with prompt tokens) focus on efficient timestep-aware fine-tuning. Compared with previous efficient timestep-aware fine-tuning methods, our TSM has a wider range of applicability (3 architectures, 2 modalities and 3 tasks, see **Table1,2 and 3** in our paper) and more advanced performance (see in **Question4**).
>
> ---
> **Question2:** The authors should support their claim by constructing some experiments to confirm compatibility or logical connections with prior timestep-based training and fine-tuning approaches.
>
> **Answer2:**  Thanks for your reminder and we added comparisons with state-of-the-art fine-tuning methods in **Question4**.  We will include these comparative experiments in the final version of the article.
>
> ---
> **Question3:** The authors should either design an experimental setup with a similar computational cost to vanilla LoRA or conduct comparative experiments against other fine-tuning methods.
>
> **Answer3:** This is an important question, and one we explore rigorously in our paper. We address this in Table 8 by comparing performance under **equal computational budgets** (e.g., `n=1, r=4, step=32k` vs. `n=4, r=4, step=8k` where `1✖️32k==4✖️8k`). Results show:
> - Training 4–8 experts outperforms single-expert setups under equal costs.
> - Performance improves with increased training steps (e.g., `step=8k` vs. `4k` for `n=1, r=4`).
>
> What's more, Table 13 further demonstrates TSM's superiority over MoE LoRA under identical budgets.
>
> ---
> **Question4:** Its superiority over existing methods remains unproven. However, the paper lacks sufficient justification for the proposed method and comparative experiments to validate its superiority. The claim would be more robust if the authors could validate that insights and observations from previous works are compatible with TSM.
>
> **Answer4:**  Thanks for your reminder. We added comparisons with state-of-the-art fine-tuning methods  and all comparative experiments use the same training and evaluation settings as the comparative methods.
>
> 1. vs. **Diffusion Model Patching via Mixture-of-Prompts, AAAI 2025. (DMP)** on LAION-5B:
> |Model|FID|
> |-|-|
> |SD1.5|47.18|
> |SD1.5+DMP|35.44|
> | SD1.5+TSM (1-stage)|33.45|
> | SD1.5+TSM (2-stage)|**32.73**|
>
> 2. vs. **Decouple-Then-Merge: Fine-Tuning Diffusion Models as Multi-Task Learning, CVPR 2025. (DeMe)** on MSCOCO:
> |Model|FID|
> |-|-|
> |SD1.5|13.42|
> |SD1.5+DeMe|13.06|
> |SD1.5+TSM (1-stage)|11.98|
> |SD1.5+TSM (2-stage)|**11.92**|
>
> ---
> **Question5:** Does TSM exhibit any scaling properties by increasing the number of intervals m?
>
> **Answer5:** This is a very important question for TSM, as we describe experimental phenomena related to scaling in our paper. We will discuss this in two aspects.
> 1. **Timestep intervals (n):** Table 8 shows optimal scaling at `n=4~8` under fixed budgets, with performance improving as training steps increase.
> 2. **Model size scaling:** Experiments in Tables 1,4,7,8,9 and 11 confirm TSM's effectiveness across model sizes (up to SD3 with 2B parameters).
>
> ---
> We sincerely appreciate your insightful comments. Please do not hesitate to let us know if you require any further information or additional explanations. If you find that our revisions have satisfactorily addressed your concerns, we would be most grateful if you could kindly consider an improved score. Thank you once again for your valuable input.

---

> > ### Comment · Reviewer_zVmL · 2025-04-02
> >
> > I appreciate the authors' detailed response and pointing out what I missed. The comprehensive ablative and comparative experiments adequately address my concerns. Thus, I have decided to increase my score.

---

> > > ### Author Response · Authors · 2025-04-02
> > >
> > > Dear Reviewer zVmL,
> > >
> > > Thank you very much for your thoughtful and perceptive evaluation. Your detailed feedback and recognition of our work’s merits truly motivate us. We are especially grateful for your discerning insight, which reflects both your deep understanding and high standards. Your positive assessment means a great deal to us.

---

### Official Review · Reviewer_whQF · 2025-03-16

**Overall Recommendation:** 3

**Summary:**

This article addresses the issue of limited model performance during the fine-tuning process of diffusion models, which arises from the use of the same LoRA across different time steps. We propose the TimeStep Master method, which employs different LoRAs at varying time step intervals to fine-tune the diffusion model. This allows different TimeStep LoRA experts to effectively capture varying noise levels. However, the article does not sufficiently elaborate on the scientific issues it addresses. For example, in the diffusion model, why is it necessary to learn different LoRAs during the fine-tuning process when the same set of U-Net network parameters can identify different noise levels? Besides, what issue can be illustrated by subfigure “a” of Figure 1? What does the hidden state of each block represent? According to the scientific problem proposed in the article, the model parameters should differ for different time steps, but the parameters are locked during the inference process. How can subfigure a be used to demonstrate the existence of this problem?

**Claims And Evidence:**

subfigure “a” of Figure 1 is not clear. How can the variance changes of the hidden states of blocks at different time steps be used to demonstrate the existence of the scientific problem that this paper aims to address?

**Essential References Not Discussed:**

Not yet

**Experimental Designs Or Analyses:**

The experimental data is sufficient, with comparisons made to the current state-of-the-art (SOTA). Ablation experiments were conducted for the two important stages of the proposed method. Additionally, generalization validation was performed for tasks related to LoRA.

**Methods And Evaluation Criteria:**

Yes, the proposed method and evaluation criteria make sense for the introduced problem.

**Other Comments Or Suggestions:**

N/A

**Other Strengths And Weaknesses:**

The article excels in the sufficiency of the experiments and the quality of writing and figures, which are significant highlights. However, it is somewhat lacking in theoretical exposition, particularly in the articulation of the scientific problem.

**Questions For Authors:**

N/A

**Relation To Broader Scientific Literature:**

From a performance perspective, it has improved the current performance of fine-tuning based on diffusion models and contributed to the advancement of LoRA. However, from a theoretical standpoint, the expansion of existing research theories is still insufficient, particularly in providing robust evidence for the hypothesis proposed in the article.

**Theoretical Claims:**

“We then hypothesis that it is the low-rank characteristic of LoRA that makes it difficult to learn complex representations at different timesteps.” The article lacks necessary evidence to support the validity of this hypothesis; simply stating that the variance of the hidden states of blocks at different time steps is large is insufficient to illustrate the problem.

---

> ### Author Rebuttal · Authors · 2025-03-28
>
> Response to Reviewer whQF:
>
> We sincerely appreciate the time and care you invested in reviewing our manuscript. Your insightful comments and suggestions have been extremely valuable, and we are grateful for the opportunity to clarify these points. Below are our detailed responses:
>
> **Question1:** What does the hidden state of each block represent?
>
> **Answer1:** In Figure 1(a), the term “hidden state” refers to the output produced by a module after the input has been processed. For example, the curve labeled “Block26” represents the variable obtained after the input has passed through the first 26 modules of the model (noting that each transformer block typically comprises both an attention module and a feed-forward network module). We hope this explanation clarifies your query.
>
> **Question2:** What issue can be illustrated by subfigure “a” of Figure 1? How can subfigure a be used to demonstrate the existence of this problem?
>
> **Answer2:** In Figure 1(a), the curves show the progression of hidden states across various timesteps within each module. This visualization reveals that even within a single module, hidden state representations vary substantially over time. This significant variation underscores the challenge of using a single set of model parameters to capture such diverse representations consistently. Our observations are in line with similar findings discussed in previous works [1–8].
>
> **Question3:** Why is it necessary to learn different LoRAs during the fine-tuning process when the same set of U-Net network parameters can identify different noise levels?
>
> **Answer3:** Addressing the challenge mentioned in **Answer2**—namely, that a single set of network parameters may struggle to adequately model the data distribution across different timesteps—it becomes a natural choice to introduce distinct parameters for various timestep intervals. Directly augmenting the full model with new parameters for each interval, however, would lead to prohibitively high computational costs. To balance efficiency and effectiveness, we employ LoRA (Low-Rank Adaptation) during fine-tuning, which allows us to efficiently adapt the network across multiple timestep intervals without incurring excessive cost.
>
> **Question4:** The model parameters should differ for different time steps, but the parameters are locked during the inference process.
>
> **Answer4:** Thank you for this excellent observation. To reconcile this, our approach incorporates a gating mechanism within the TSM 2-stage framework. Although the underlying network parameters remain fixed during the inference process, the gating mechanism dynamically adjusts the contribution (or weights) of multiple experts. This adaptive activation allows the model to effectively tailor its response to different timesteps. We have provided a detailed demonstration of the effectiveness of this experts ensemble using the 2-stage gating approach in Tables 4, 5, 6, and 11, and we further analyze the gating structure in Table 7.
>
> **References**
> [1] Multi-Architecture Multi-Expert Diffusion Models, AAAI 2024.
> [2] eDiff-I: Text-to-Image Diffusion Models with an Ensemble of Expert Denoisers
> [3] Towards Practical Plug-and-Play Diffusion Models, CVPR 2023.
> [4] Mixture of efficient diffusion experts through automatic interval and sub-network selection, ECCV 2024.
> [5] Improving training efficiency of diffusion models via multi-stage framework and tailored multi-decoder architecture, CVPR 2024.
> [6] Switch Diffusion Transformer: Synergizing Denoising Tasks with Sparse Mixture-of-Experts, ECCV 2024.
> [7] Denoising Task Routing for Diffusion Models, ICLR 2024.
> [8] Addressing Negative Transfer in Diffusion Models, NeurIPS 2023.
>
> ---
> Thank you once again for your thoughtful and constructive feedback. Your insightful suggestions have greatly enhanced our manuscript, and we have carefully integrated the corresponding experimental details and clarifications into the final version. We sincerely hope these revisions have addressed your concerns, and if you feel they have been satisfactorily resolved, we would be most grateful if you could consider revising your evaluation score accordingly. Please do not hesitate to let us know if you need any further explanations.

---

### Official Review · Reviewer_4dgm · 2025-03-17

**Overall Recommendation:** 4

**Summary:**

This paper introduces TimeStep Master (TSM), a method that employs multiple LoRA experts, each specialized in specific timestep regions. The authors empirically analyze the degradation caused by sharing LoRA parameters across all timesteps, which motivates their proposal of expert LoRAs tailored to distinct timestep ranges. Specifically, they construct specialized LoRAs by evenly dividing timesteps (or noise levels) and further refining them through multi-scale partitioning to capture coarse-to-fine timestep variations. These LoRA parameters are then dynamically routed in a mixture-of-experts (MoE) fashion, enabling effective ensembling.

**Claims And Evidence:**

I think most claims are clear, as the effectiveness of using specialized parameters for divided timesteps has already been demonstrated.

**Essential References Not Discussed:**

As mentioned in Relation to Broader Scientific Literature, this work is closely related to diffusion timestep division. Therefore, it would be beneficial for the paper to explicitly discuss this topic.

Multi-Experts [2, 3, 4, 5, 6]: These approaches employ specialized parameters to mitigate conflicts between denoising tasks across different timesteps, as discussed in [1].
Mixture-of-Experts (MoE) [7, 8, 9]: This technique is used in their method specifically for ensembling LoRA experts.
Given these connections, it would be useful to compare the proposed approach with these existing methods. The key novelty of this work lies in multi-scale timestep division, which enables the model to leverage both broadly trained LoRA experts that capture overall denoising patterns and fine-grained LoRA experts that specialize in specific timesteps. This flexibility makes the method more adaptable to different granularity levels of denoising tasks.

As long as the additional computational cost remains manageable, I believe this is a promising approach.

### Reference
- [1] Addressing Negative Transfer in Diffusion Models, Neurips 2023,
- [2] Multi-Architecture Multi-Expert Diffusion Models, AAAI 2024
- [3] eDiff-I: Text-to-Image Diffusion Models with an Ensemble of Expert Denoisers
- [4] Towards Practical Plug-and-Play Diffusion Models, CVPR 2023
- [5] Mixture of efficient diffusion experts through automatic interval and sub-network selection, ECCV 2024
- [6] Improving training efficiency of diffusion models via multi-stage framework and tailored multi-decoder architecture, CVPR 2024.
- [7] Switch Diffusion Transformer: Synergizing Denoising Tasks with Sparse Mixture-of-Experts, ECCV 2024
- [8] Scaling Diffusion Transformers to 16 Billion Parameters,
- [9] EC-DIT: Scaling Diffusion Transformers with Adaptive Expert-Choice Routing, ICLR 2025.

**Experimental Designs Or Analyses:**

I think the training costs reported in Table 3 are somewhat exaggerated, but it is not a critical issue.

**Methods And Evaluation Criteria:**

The evaluation metrics and datasets appear reasonable, but the comparison is primarily against LoRA, leaving out several baselines. I believe their setup should include comparisons with multi-expert methods and MoE-like diffusion models.

**Other Comments Or Suggestions:**

N/A

**Other Strengths And Weaknesses:**

## Strength

1. The paper is clearly written and well-structured, making it easy to follow the motivation, methodology, and results.

2. The authors conduct a thorough experimental evaluation, demonstrating the effectiveness of their approach with various settings.

3. The concept of multi-scale timestep division for LoRA is an interesting and novel aspect.

## Weakness
1. Lack of discussions with timestep division in the diffusion model literature
   - This paper considers LoRA specialization based on timestep division as its core idea. However, the discussion on similar approaches in the diffusion model literature feels somewhat lacking.

   - Multi-expert methods [2, 3, 4, 5, 6] have proposed a conceptually similar approach by assigning specialized parameters to different timestep regions, thereby resolving conflicts between denoising tasks across timesteps. These works have demonstrated the effectiveness of such strategies [1], making them highly relevant to this paper’s methodology. Given the strong similarity between these approaches and the timestep-specialized LoRA concept, a more detailed discussion would be beneficial.

   - Furthermore, prior works have already explored timestep division for fine-tuning diffusion models in setups very similar to the one proposed in this paper. Notably, DMP (Diffusion Model Patching via Mixture-of-Prompts) [7] and Decouple-Then-Merge [8] explicitly adopt a strategy of dividing timesteps and merging them later, making them particularly relevant. Given the conceptual overlap, an empirical comparison with these works would provide a clearer positioning of this paper’s contributions.

2. Additional Training Costs for Multi-Expert Convergence

   - A major drawback of multi-expert methods is the increased training cost due to the need for each expert to converge on its respective timestep region. Since TimeStep Master (TSM) follows a similar paradigm by specializing LoRA experts for different timesteps, it inherently inherits this issue. However, the paper does not provide sufficient validation on how well the model mitigates this problem. To address this, it would be beneficial to include an analysis of training efficiency. One possible way to validate this is by presenting a training step vs. evaluation metric curve, similar to iteration-based learning curves, to show the additional computational overhead and how performance scales with training.


### Reference
- [1] Addressing Negative Transfer in Diffusion Models, NeurIPS 2023
- [2] Multi-Architecture Multi-Expert Diffusion Models, AAAI 2024
- [3] eDiff-I: Text-to-Image Diffusion Models with an Ensemble of Expert Denoisers
- [4] Towards Practical Plug-and-Play Diffusion Models, CVPR 2023
- [5] Mixture of Efficient Diffusion Experts through Automatic Interval and Sub-Network Selection, ECCV 2024
- [6] Improving Training Efficiency of Diffusion Models via Multi-Stage Framework and Tailored Multi-Decoder Architecture, CVPR 2024
- [7] Diffusion Model Patching via Mixture-of-Prompts (DMP), AAAI 2025
- [8] Decouple-Then-Merge: Fine-Tuning Diffusion Models as Multi-Task Learning, CVPR 2025

**Questions For Authors:**

N/A

**Relation To Broader Scientific Literature:**

In this paper, the authors primarily focus on enhancing LoRA for efficient fine-tuning of diffusion models. Previous works on leveraging LoRA for efficient tuning have explored three main directions: (1) dataset design, (2) distillation objectives (e.g., reinforcement learning), and (3) LoRA architecture. This work falls under the third category, LoRA architecture, aiming to improve it by incorporating timestep division, MoE, multi-expert modeling, and multi-task learning—concepts that are extensively studied in the diffusion model literature.

Notably, some closely related works that explore similar approaches but are not cited include DMP and Decouple-Then-Merge (but not cited):
- Diffusion Model Patching via Mixture-of-Prompts (DMP), AAAI 2025
- Decouple-Then-Merge: Fine-Tuning Diffusion Models as Multi-Task Learning, CVPR 2025
These works share similarities with the proposed method and should be considered in the discussion.

**Theoretical Claims:**

N/A

---

> ### Author Rebuttal · Authors · 2025-03-28
>
> Response to Reviewer 4dgm:
>
> Thank you for your thorough review of our paper and your valuable suggestions. Below are our point-by-point responses:
>
> **Question1:** Some closely related works that explore similar approaches but are not cited include DMP and Decouple-Then-Merge.
>
> **Answer1:** We sincerely appreciate your reminder.
> 1. We were previously unaware of the DMP work, which shares similarities with our method. We will elaborate on this comparison in **Question4**.
> 2. Decouple-Then-Merge was published in CVPR 2025, with its release date preceding the ICML 2025 submission deadline. Unfortunately, we could not access this work during our submission period. However, we will elaborate on this comparison in **Question4** and we will include citations to both works in the final version.
> ---
> **Question2:** Essential References Not Discussed: Multi-Experts [2,3,4,5,6]: These approaches employ specialized parameters to mitigate conflicts between denoising tasks across different timesteps, as discussed in [1]. Mixture-of-Experts (MoE) [7,8,9]: This technique is used in their method specifically for ensembling LoRA experts.
>
> **Answer2:** Thank you for highlighting the need for broader discussion.  We deeply appreciate your additional references.
> We will discuss the differences between us in detail below and these will be incorporated into the final version.
>
> |Method|Limitations|TSM Advantages|
> |-|-|-|
> |[2] Multi-Architecture Multi-Expert Diffusion Models. [3] eDiff-I: Text-to-Image Diffusion Models with an Ensemble of Expert Denoisers. [6] Improving training efficiency of diffusion models via multi-stage framework and tailored multi-decoder architecture.|Require training multiple large models (storage/inference challenges)|Uses LoRA for **efficient** timestep interval tuning in 1-stage and ensembles knowledge in 2-stage|
> |[4] Towards Practical Plug-and-Play Diffusion Models.|Applies LoRA only to guidance models|Directly integrates LoRA into both text encoder and diffusion models|
> |[5] Mixture of efficient diffusion experts through automatic interval and sub-network selection|Focuses on timestep redundancy for inference speed|Achieves **superior performance** via two-stage specialization and ensembling|
> |[7] Switch Diffusion Transformer: Synergizing Denoising Tasks with Sparse Mixture-of-Experts. [8] Scaling Diffusion Transformers to 16 Billion Parameters. [9] EC-DIT: Scaling Diffusion Transformers with Adaptive Expert-Choice Routing.|Need to train the model from scratch, which consumes resources.|TSM is compatible with the existing diffusion model and continues to improve model performance.|
> ---
> **Question3:** The discussion on similar approaches in the diffusion model literature feels somewhat lacking. Multi-expert methods [2,3,4,5,6] have proposed a conceptually similar approach by assigning specialized parameters to different timestep regions, thereby resolving conflicts between denoising tasks across timesteps.
>
> **Answer3:** Thank you for your extensive research on our article. We have supplemented **Question2** with detailed discussions and will include them in the final version.
>
> ---
> **Question4:** DMP [7] and Decouple-Then-Merge [8] explicitly adopt a strategy of dividing timesteps and merging them later, making them particularly relevant.
>
> **Answer4:** We give a detailed comparison below and all experiments use the same training and evaluation settings
> 1. DMP
> - Trains gating modules and prompt tokens simultaneously
> - TSM uses **Two-Stage training**:
>   - Stage 1: Specializes experts per timestep interval
>   - Stage 2: Freezes experts and learns gating modules
>
> LAION-5B Results (FID↓):
> | Model|FID|
> |-|-|
> | SD 1.5|47.18|
> | SD1.5 + DMP|35.44|
> | SD1.5 + TSM (Stage1)|33.45|
> | SD1.5 + TSM (Stage2)|**32.73**|
>
> 2. Decouple-Then-Merge (DeMe)
> - Merges parameters via weighted averaging
> - TSM uses **gating mechanisms** (See Table 7 for gating mechanism ablation) to preserve timestep-specific knowledge
>
> MSCOCO Results (FID↓):
> | Model|FID|
> |-|-|
> | SD 1.5|13.42|
> | SD1.5 + DeMe|13.06|
> | SD1.5 + TSM (1-stage)|11.98|
> | SD1.5 + TSM (2-stage)|**11.92**|
>
> ---
> **Question5:** Additional Training Costs for Multi-Expert Convergence. A major drawback of multi-expert methods is the increased training cost. Since TimeStep Master (TSM) follows a similar paradigm by specializing LoRA experts for different timesteps, it inherently inherits this issue.
>
> **Answer5:** We rigorously **evaluated computational efficiency** in **Table 8**:
> - **Equal Cost Settings**: Training multi-experts (e.g., `n=4, r=4, step=8k`) outperforms single-expert training (`n=1, r=4, step=32k`) under the same training cost (4✖️8k==1✖️32k)
> - **Scaling Benefits**: Performance improves consistently with more training steps (e.g., `step=8k` vs. `step=4k` for `n=1, r=4`)
> ---
> We sincerely thank you again for your insightful feedback. All suggestions and experimental results will be integrated into the final version. Please let us know if further clarifications are needed.

---

> > ### Comment · Reviewer_4dgm · 2025-04-02
> >
> > The authors' rebuttal well addressed my concerns, so I raised my score.

---

> > > ### Author Response · Authors · 2025-04-02
> > >
> > > Dear Reviewer 4dgm,
> > >
> > > Thank you very much for your thoughtful and constructive comments. We truly appreciate your keen insights and the careful consideration you gave to our work. Your perspective has been invaluable in helping us refine our paper, and we are grateful for your clear vision and support.

---

### Decision · Program_Chairs · 2025-05-01

**Decision:**

Accept (poster)

**Comment:**

This paper introduces TimeStep Master (TSM), which is a two-stage fine-tuning approach for diffusion models using LoRA. TSM applies different LoRA adapters at different timesteps and combines them through a core-context mixture strategy. This improves noise handling and adaptability, achieving state-of-the-art results across tasks like domain adaptation, post-pretraining, and model distillation on various model types and data modalities. Four reviewers checked this paper and they have slightly different opinions about this work in the final recommendations. Notably, one reviewer raised the score after rebuttal and believes this work is good enough to be accepted. Based on the comments, recommendations, and post-rebuttal discussion, the authors’ point-to-point rebuttal addressed most of these concerns so that no significant issues remain. This paper is acceptable in its current form.